# *A Tale of Pronouns:* Interpretability Informs Gender Bias Mitigation for Fairer Instruction-Tuned Machine Translation

**Giuseppe Attanasio[♡], Flor Miriam Plaza-del-Arco[♡], Debora Nozza[♡], Anne Lauscher[♣]**

[♡] Bocconi University, Milan, Italy
[♣] University of Hamburg, Hamburg, Germany
giuseppe.attanasio3@unibocconi.it

## Abstract

Recent instruction fine-tuned models can solve multiple NLP tasks when prompted to do so, with machine translation (MT) being a prominent use case. However, current research often focuses on standard performance benchmarks, leaving compelling fairness and ethical considerations behind. In MT, this might lead to misgendered translations, resulting, among other harms, in the perpetuation of stereotypes and prejudices. In this work, we address this gap by investigating whether and to what extent such models exhibit gender bias in machine translation and how we can mitigate it. Concretely, we compute established gender bias metrics on the WinoMT corpus from English to German and Spanish. We discover that IFT models default to male-inflected translations, even disregarding female occupational stereotypes. Next, using interpretability methods, we unveil that models systematically overlook the pronoun indicating the gender of a target occupation in misgendered translations. Finally, based on this finding, we propose an easy-to-implement and effective bias mitigation solution based on few-shot learning that leads to significantly fairer translations.[1]

## 1 Introduction

Instruction fine-tuned (IFT) models, such as Flan-T5 (Chung et al., 2022) and mT0 (Muennighoff et al., 2023a), are trained on large corpora of machine learning tasks verbalized in natural language and learned through standard language modeling. The large and diverse mixture of training tasks has led to unmatched transfer performance – if prompted properly, models are able to virtually solve any standard NLP task, including sentiment analysis, natural language inference, question answering, and more (Sanh et al., 2022).

However, most efforts on their evaluation have focused on standard benchmarks only, with

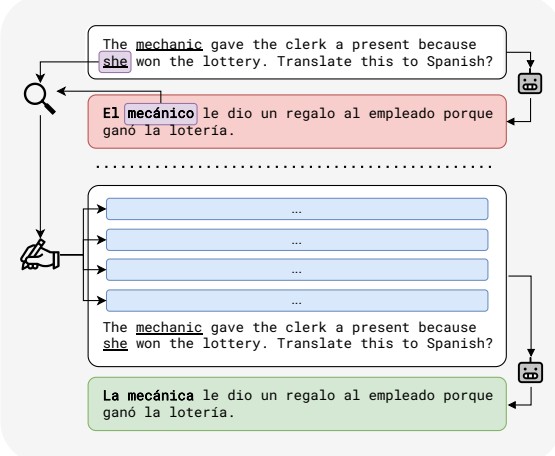

Figure 1: Model translation suffering from occupational gender bias (top); coreferents underlined. Our interpretability analysis on pronouns and professions (purple boxes) informs the selection of debiasing examples. Human-translated demonstrations (blue) enable fairer translations via few-shot learning ("**La** mecán**ica**", correct feminine inflection for "the mechanic").

a prominent focus on testing zero-shot abilities (Chung et al., 2022) and cross-lingual generalization (Muennighoff et al., 2023b), and have thus largely ignored the models' social impact (Hovy and Spruit, 2016). This lacuna is extremely surprising as (a) IFT models are based on pretrained language models, which are widely known to encode societal biases and unfair stereotypes (Nadeem et al., 2021; Nozza et al., 2021, *inter alia*); and (b) exposing models to many fine-tuning sources can exacerbate biased behaviors as stereotypical demonstrations add up (Srivastava et al., 2022).[2] As a result, we expect instruction-tuned models to encode societal biases and unfair stereotypes, possibly even beyond the extent of their base models. Still, few efforts have been spent on bias evaluation and mitigation for these models so far (a notable

---

[1]Code and data artifacts at https://github.com/MilaNLProc/interpretability-mt-gender-bias.

[2]For a reference, FLAN models are trained on a mixture of 1.8K tasks (Chung et al., 2022).

exception being provided by Akyürek et al. (2022)), putting their societal beneficial use at risk.

In this work, we address this research gap by studying occupational gender bias in zero- and few-shot setups in one of the, arguably, most prominent NLP applications to date, machine translation (MT). To this end, we use the established WinoMT benchmark (Stanovsky et al., 2019) and study the translation from English to Spanish and German, two morphologically diverse languages that both require inflecting multiple syntactic items. We experiment with Flan-T5 and mT0, two state-of-the-art IFT models, controlling for several factors such as the prompt template, model size, and decoding strategy. Importantly, we make use of established interpretability tools to shed light on *when* and *how* such models use lexical clues when picking the right (or wrong) gender inflection for a target profession. We then use those insights for informing an easy-to-use and effective bias mitigation approach.

**Contributions and Findings.** Our contributions are three-fold: (**1**) *we provide one the few studies on bias in instruction-tuned models to-date*. Focusing on the example of MT and gender bias, we show that despite getting better at zero-shot translation, such models default to male-inflected translations, even in the presence of overt female pronouns and disregarding female occupational stereotypes. (**2**) To our knowledge, *we are among the first to acknowledge the potential of interpretability methods to study IFT language models and why they produce biased predictions*. Based on attribution interpretability, we find that models systematically ignore the pronoun (and thus, the conveyed gender information) when producing misgendered translations. In contrast, correctly translated professions relate to higher contributions of the pronoun in the choices taken. (**3**) *Based on our insights, we propose a novel and easy-to-use bias mitigation method – informed by interpretability scores!* The differences in the attribution scores lead us to hypothesize that models that are used in a few-shot setup would benefit from provided translations mostly, if exactly in those examples they would normally overlook the pronoun. We hence propose a few-shot learning-based debiasing approach, in which we use interpretability scores to select the in-context exemplars. Figure 1 shows an example of the resulting approach. The solution is simple-yet-effective, leading to significantly fairer translations

with as few as four human-translated exemplars.

Overall, **our findings prove interpretability as a valuable tool for studying and mitigating bias in language models, both as a diagnostic tool and a signal driving bias mitigation approaches**. We release code and data artifacts hoping to foster future research in this direction.

## 2 Experimental Setup

The primary use case for instruction-tuned models is to tackle standard NLP tasks by formulating a specific request in the input prompt. Here, we experiment with MT, triggered by a specific phrasing such as "`Translate this into Spanish.`"

In particular, we set to study whether such models exhibit gender bias concerning occupations. While doing so, we apply established interpretability metrics to explain why the model preferred specific gender inflections. Later (§4), we propose a novel debiasing approach based on few-shot learning informed by the interpretability findings.

### 2.1 Gender Bias in Machine Translation

**Bias Statement.** We expect LMs to inflect gender in occupation words according to overt contextual and lexical clues. Instead, a biased model is one, which relies on stereotypical gender-role associations. Both open source and commercial MT systems have been shown to rely on these associations, with a marked tendency to associate women with less prestigious roles (e.g., Stanovsky et al., 2019; Saunders and Byrne, 2020; Chung et al., 2022, *inter alia*). Echoing Blodgett et al. (2020), such systems risk *representational* harms, as they portray women in a less favorable light than men.

**Dataset.** We base our experiments on WinoMT (Stanovsky et al., 2019), a well-known benchmark for evaluating gender bias in MT.

The collection is based on templates. Each instance mentions two professions and a pronoun coreferent to one of them (see Figure 1 for an example). When translating from English, a notional gender language, to Spanish or German, two grammatical gender languages, the pronoun dictates the coreferent inflection because of syntactic agreement. For example, the sentence in Figure 1 with "she" as the leading pronoun should translate to "La

mecánica" (eng: the female mechanic).[3] The task is challenging as many of the occupations found in WinoMT have stereotypical gender-role associations in society (e.g., nurse to women, developer to men). Indeed, the WinoMT corpus distinguishes between stereotypical and anti-stereotypical templates, which permits us to derive more insights.

**Evaluation Metrics.** We evaluate gender bias using the measures proposed along with WinoMT Stanovsky et al. (2019). The authors conceptualize bias as differences in group performance indicated by $\Delta_G$ and $\Delta_S$. $\Delta_G$ corresponds to the difference in performance ($F_1$ macro) between translations with a female and male referent. $\Delta_S$ measures the difference in performance between stereotypical and anti-stereotypical examples, as per US Labour statistics in Zhao et al. (2018). Additionally, we also report the overall accuracy.

## 2.2 Interpretability for Gender Bias in MT

**Interpretability for Translations.** For every translated instance, we compute and collect word attribution interpretability scores from target to source tokens. Word attribution scores (i.e., *saliency* scores), measure each input token's contribution to the choice of any translation token.

We compute word attributions as follows: first, we extract raw attribution scores using Integrated Gradients (IG; Sundararajan et al., 2017), a commonly used feature attribution algorithm. Let $A_r = \mathbb{R}^{S_r \times T_r \times h}$ be the output of IG, a matrix of attribution scores, where $S_r$ and $S_t$ are the number of source and target tokens, respectively, and $h$ is the hidden dimension of the input embeddings.

Next, we aggregate scores applying two consecutive functions, i.e., $A = g(f(A_r))$, where $f : \mathbb{R}^{S_r \times T_r \times h} \rightarrow \mathbb{R}^{S \times T \times h}$ and $g : \mathbb{R}^{S \times T \times h} \rightarrow \mathbb{R}^{S \times T}$ aggregate raw scores over each word's sub-tokens and the hidden dimension, respectively. $S$ and $T$ are the number of source and target words as split by whitespaces, respectively. We set $f$ to take the highest absolute value in the span, preserving the sign, and $g$ to the Euclidean norm. We provide more details in Appendix B.1. As a result, each item $a_{i,j} \in A$ will reflect the contribution that source word $i$ had in choosing the target word $j$.

**Interpretability Signals for Gender Bias.** Word attribution scores provide a clear, measurable quantity to inspect and debug machine translation models. We, therefore, study such scores and relate recurring patterns to misgendered translations.

We extracted several word attribution scores. First, we observe the "alignment" importance between translations $a_{prof,prof}$, the importance of the English profession word for the target profession (mechanic and mecánico in Figure 1). Then, we also report a control attribution score $a_{ctrl,prof}$ as the importance of the first source token ("The") toward the target profession.

The most promising aspect we study is the contribution score $a_{pron,prof}$, i.e. the importance the source pronoun has in choosing the final form of the target profession (she and mecánico in Figure 1). Intuitively, the models need to use this overt lexical clue to choose the correct inflection. Note that extracting $a_{prof,pron}$ requires aligning the source and target sentences because WinoMT does not release the profession position within the sentence, while the pronoun position is given. We tested both Fast Align (Dyer et al., 2013) and a custom dictionary matching approach and proceeded with the latter because of better quality. In particular, we prompted GPT-3.5 (Ouyang et al., 2022, gpt-3.5-turbo, accessed in early June 2023) to translate all the target professions in WinoMT into the masculine and feminine inflected forms in Spanish and German.[4] After checking and fixing any grammatical errors, we perform hard string matching of the MT output against the word translations.

## 2.3 Instruction Fine-Tuned Models

**Base Models.** We study variants of two recently introduced IFT models.[5]

*Flan-T5* (Chung et al., 2022) is a sequence-to-sequence language model based on the T5 architecture (Raffel et al., 2020). The model has been pre-trained with standard language modeling objectives and subsequently fine-tuned on the FLAN collection (Longpre et al., 2023), counting more than 1,800 NLP tasks in over 60 languages. We test the 80M (Small), 250M (Base), 780M (Large),

---

[3] WinoMT includes a small set of examples using neutral pronouns to test gender-neutral translation (GNT). As GNT is a developing field, we report preliminary insights on GNT cases (§5). Otherwise, we restrict the rest of the work to binary gender.

[4] This dictionary contains a single entry pair per profession. Hence, we do not match professions against multiple correct translations (e.g., we match the Spanish "maestro/maestra" but not "profesor/profesora" for "teacher").

[5] We follow the nomenclature from Chung et al. (2022) and refer to these models as *instruction fine-tuned* models, in favor of alternatives such as *multi-task prompted* models. See Appendix A for more details.

| Model | C-22 | C-20 | BERTScore |
|-------|------|------|-----------|
| Marian NMT | **0.85** | **0.42** | **0.89** |
| Flan-T5 | 0.78 | 0.18 | 0.86 |
| mT0 | 0.80 | 0.27 | 0.82 |
| Marian NMT | **0.80** | **0.43** | **0.85** |
| Flan-T5 | 0.75 | 0.31 | 0.82 |
| mT0 | 0.75 | 0.42 | 0.81 |

Table 1: COMET-22, COMET-20, and BERTScore performance on Europarl in En-Es (top) and En-De (bottom). Best results per language in bold.

3B (XL), and 11B (XXL) model sizes.

*mT0* (Muennighoff et al., 2023b) is a mT5 model (Xue et al., 2020) fine-tuned on xP3, covering 13 tasks across 46 languages with English prompts. We test the 300M (Small), 580M (Base), 1.2B (Large), 3.7B (XL), and 13B (XXL) model sizes.

Both model types have been fine-tuned verbalizing NLP tasks into a text-to-text format and using standard encoder-decoder language modeling loss. Moreover, FLAN and xP3 training mixtures both contain machine translation tasks. For all open models, we use the Hugging Face Transformers library (Wolf et al., 2020).

**Model Configuration and Tuning.** We consider two standard prompt templates and five decoding strategies to account for possible variations with instruction-tuned models. See Appendix C.2 for details. In order to assess the translation quality and select the best instruction-tuned model configuration, we perform an extensive evaluation within a benchmark evaluation framework.

We use the state-of-the-art Europarl corpus (Koehn, 2005) to evaluate zero-shot translation quality.[6] We use the benchmark evaluation metrics COMET (reference-based -22 (Rei et al., 2022) and reference-free -20 (Rei et al., 2020)) and BERTScore (Zhang et al., 2019). We also include BLEU (-2 and -4) (Papineni et al., 2002) for comparison with prior work.

## 3 Results

### 3.1 General Translation Quality

The results on EuroParl (Appendix C.2) show that the best overall quality is obtained with beam search decoding (n=4, no sampling) and the prompt

template "{src_text} Translate this to {tgt_lang}?". Most importantly, we found that model size is key to enabling zero-shot translation (see Table 9). This crucial finding suggests that **smaller models (i.e., <11B) do not yield translations of sufficient quality** and their usage in a zero-shot setup can be problematic. We will focus on the largest models (XXL variants) for the rest of the paper and simply refer to them as Flan-T5 and mT0 for conciseness.

Table 1 reports the zero-shot performance of Flan-T5 and mT0 compared to supervised baseline Marian NMT models (Junczys-Dowmunt et al., 2018)[7]. Flan-T5 and mT0 slightly underperform supervised baselines. However, they show competitive zero-shot performance as measured by COMET-22 and BERTScore. COMET-20 quality estimation metric show less encouraging results, especially for Flan-T5 (see Table 9 for a full breakdown). Overall, these results suggest that zero-shot translation with instruction-tuned models is almost as valid as specialized supervised models, further motivating their adoption in real use cases.

### 3.2 Gender Bias in Instruction-Tuned Models

Table 2 reports the results on WinoMT gender bias metrics. We report several interesting findings. Generally, **Flan-T5 is competitive**. For both languages, it significantly outperforms mT0 in terms of accuracy and bias evaluation. Moreover, considering commercial systems reported in Stanovsky et al. (2019), GPT-3.5 (Ouyang et al., 2022, gpt-3.5-turbo, accessed in early June 2023), and our supervised baseline, Flan-T5 achieves the best accuracy and $\Delta_G$ in En-Es, and the best $\Delta_S$ in En-De. However, it falls severely short on $\Delta_S$ En-Es, where the supervised Marian NMT model tops the board. We addressed this weakness using few-shot learning (§4).

As for negative findings, we see that **mT0 retains high occupation biases**. It is the second worst system for accuracy, $\Delta_G$ and $\Delta_S$ in En-Es, with similar results in En-De.[8] Notably, zero-shot translations from GPT-3.5 are biased and worse than supervised baselines and instruction-tuned Flan-T5. Despite the interesting finding, we do not explore GPT-3.5 further since the attribution

---

[6]We use the WMT' 06 test splits at https://www.statmt.org/wmt06/shared-task/

[7]En-Es: https://huggingface.co/Helsinki-NLP/opus-mt-en-es, En-De: https://huggingface.co/Helsinki-NLP/opus-mt-en-de

[8]This negative finding is even more significant considering that systems from Stanovsky et al. (2019) date back to 2019.

| Model | Spanish | | | German | | |
|---|---|---|---|---|---|---|
| | Acc | $\Delta_G$ | $\Delta_S$ | Acc | $\Delta_G$ | $\Delta_S$ |
| Google Translate* | 53.1 | 23.4 | 21.3 | 59.4 | 12.5 | 12.5 |
| Microsoft Translator* | 47.3 | 36.8 | 23.2 | **74.1** | **0.0** | 30.2 |
| Amazon Translate* | 59.4 | 15.4 | 22.3 | 62.4 | 12.0 | 16.7 |
| Marian NMT | 56.8 | 16.9 | **19.7** | 62.0 | 9.9 | 15.2 |
| GPT-3.5 | 55.2 | 23.1 | 48.5 | 48.3 | 25.2 | 24.6 |
| Flan-T5-XXL | **65.1** | **7.2** | 35.1 | 66.9 | 2.7 | **-0.2** |
| mT0-XXL | 52.5 | 27.8 | 42.7 | 56.3 | 26.1 | 25.6 |

Table 2: Gender bias evaluation on WinoMT. *Results reported from Stanovsky et al. (2019).

techniques require access to the model weights.

Overall, these findings suggest that instruction-tuned models can reasonably solve the task in a zero-shot setup, with Flan models being superior to mT0.

### 3.3 Inspecting Word Attribution Scores

Word attribution scores give us additional insights into the model's biased behavior. Table 3 shows the average word attribution scores introduced in Section 2.2 grouped by model, language, gender, and stereotypical and anti-stereotypical cases. The table also provides disaggregated accuracy for better understanding. Using our dictionary-based string matching, we found the target profession (i.e., inflected in either of the two forms) in 64% (En-Es) and 39% (En-De) for Flan-T5 and 70% and 49% for mT0.[9]

Male cases are always associated with the highest accuracy, with a difference of 21% and 62% between male and female cases for Flan-T5 and mT0, respectively. Moreover, stereotypical male cases hold the highest performance across all groups. This finding highlights (1) a strong tendency to default to masculine forms and (2) that male stereotypical cases are easier to translate on average. These results confirm those obtained by observing $\Delta_G$ and $\Delta_S$ in the previous paragraph.

In three out of four cases, stereotypical male cases also hold the highest $a_{prof,prof}$ value. However, while the most accurate cases are those in which the source-target profession attribution is the strongest, the opposite is not true (mT0, En-Es, stereo, M has not the highest $a_{prof,prof}$). Therefore, we conclude there is no clear correlation between

accuracy and $a_{prof,prof}$.

More insightful findings can be derived by the word attribution score $a_{pron,prof}$, i.e., the source pronoun importance for translating the gendered profession. Intuitively, *source pronoun should be the model's primary source of information for selecting the correct gender inflection.* If we observe low values for this score, we can assume the model has ignored the pronoun for translating. This pattern is especially true for stereotypical male cases: despite their high accuracy, $a_{pron,prof}$ scores are low. We observed an opposite trend for stereotypical female cases, where $a_{pron,prof}$ scores are the highest, but accuracy is low. Interestingly, $a_{pron,prof}$ is highly asymmetrical between female and male cases. In six out of eight (model, language, and stereotype) groups, $a_{pron,prof}$ is higher for females than males. Regarding stereotypical vs. anti-stereotypical occupations, $a_{pron,prof}$ is higher for the latter on three out of four model-language pairs. This statistic supports the intuition that anti-stereotypical cases are where the model is most challenged, particularly for female professions, which consistently have the lowest accuracy. These findings, taken together, reveal a concerning bias in the way professions are portrayed in the models. Even after making an extra effort to consider pronouns, **professions are frequently translated into their male inflection, even when they would be stereotypically associated with the female gender.**[10]

Finally, we studied how $a_{prof,prof}$ and $a_{pron,prof}$ relate to translation errors. Specifically, we computed the average $a_{prof,prof}$ and $a_{pron,prof}$ across all correctly and non-correctly translated examples and measured their relative difference. Ta-

---

[9]Manual inspection revealed that fewer matches in En-De are due to the model's frequent use of synonyms, English profession words, or wrong translation.

[10]The word attribution score created as a control variable, $a_{ctrl,prof}$, does not show any clear trend as expected.

| Model | Lang | Stereotypical | Gender | $a_{ctrl,prof}$ | $a_{prof,prof}$ | $a_{pron,prof}$ | Acc |
|---|---|---|---|---|---|---|---|
| Flan-T5 | Es | Anti | F | **0.192** | 0.163 | **0.174** | 39.49 |
| | | | M | 0.142 | 0.166 | 0.150 | 79.97 |
| | | Stereo | F | 0.155 | 0.157 | 0.161 | 76.39 |
| | | | M | 0.187 | **0.167** | 0.162 | **94.95** |
| | De | Anti | F | 0.244 | 0.145 | 0.254 | 68.10 |
| | | | M | 0.209 | 0.133 | 0.221 | 77.08 |
| | | Stereo | F | **0.231** | 0.129 | **0.266** | 64.65 |
| | | | M | 0.218 | **0.164** | 0.149 | **80.18** |
| mT0 | Es | Anti | F | **0.240** | 0.132 | 0.201 | 11.52 |
| | | | M | 0.201 | **0.149** | 0.177 | 86.02 |
| | | Stereo | F | 0.203 | 0.143 | **0.203** | 36.62 |
| | | | M | 0.236 | 0.131 | 0.189 | **94.32** |
| | De | Anti | F | **0.230** | 0.137 | 0.199 | 21.65 |
| | | | M | 0.220 | 0.124 | 0.210 | 86.15 |
| | | Stereo | F | **0.230** | 0.121 | **0.244** | 38.89 |
| | | | M | 0.219 | **0.141** | 0.165 | **91.92** |

Table 3: Word attribution scores and accuracy (%) disaggregated by model, language, stereotypical and anti-stereotypical, and expected gender inflection. Highest values in each (model, language) group in bold.

| Stereotypical | Gender | $\Delta_{prof,prof}$ | $\Delta_{pron,prof}$ |
|---|---|---|---|
| Anti | F | +5.70 | -14.23 |
| | M | -3.46 | +5.03 |
| Stereo | F | +2.36 | -13.80 |
| | M | -0.04 | -2.58 |
| Average | | +1.14 | -6.40 |

Table 4: Relative difference (%) between correct and wrong translations for mean $a_{prof,prof}$ and $a_{pron,prof}$. Results averaged over models and languages and disaggregated by stereotypical setup and expected gender inflection. Positive values indicate that the score is higher in correct translations.

ble 4 reports the results. $a_{prof,prof}$ does not show any clear associations with errors, and it is hard to relate it to biased behaviors. $a_{pron,prof}$, on the other hand, shows again high asymmetry between female and male cases. Interestingly, **models attend to the source pronoun sensibly less when wrongly translating female referents** (-14% in both anti-stereotypical and stereotypical cases), but the same is not valid for male cases.

All these results support the use of ad-hoc interpretability methods for discovering word attribution scores associations with desirable (or undesirable) behavior, thereby serving as proxies for subsequent interventions.

## 4 Interpretability-Guided Debiasing

Taking stock of the findings in Section 3, we know that models overtly ignore gender-marking pronouns but also that interpretability scores provide us with a reliable proxy for the phenomenon.

Therefore, we hypothesize we can reduce the model's errors and, in turn, its translation bias by "showing" examples where it would typically overlook the pronoun, each accompanied by a correct translation. Building on recent evidence that large models can solve tasks via in-context learning (Brown et al., 2020b), we implement this intuition via few-shot prompting. Crucially, we use interpretability scores to select in-context exemplars.

We proceed as follows. First, we extract examples with lowest $a_{pron,prof}$ importance score, i.e., instances where the model relied the least on the gender-marking pronoun to inflect the profession word. Then, we sample N exemplars from this initial pool and let them be translated by humans. Finally, we use these exemplars as few-shot seeds, simply prepending them to the prompt. Figure 1 shows as end-to-end example of the process.

We experiment with N=4, sampling directly from WinoMT, and stratifying on stereotypical/anti-stereotypical and male/female groups to increase coverage. We translate the seeds ourselves.[11] As templates contain one more profession whose gender is unknown (here, NT: non-target), we experiment with either inflecting it to its feminine form (NT-Female), its masculine form (NT-Male), or a randomly choosing between the two (NT-Random). See Appendix D.1 for full details on the few-shot prompt construction.

As a baseline debiasing approach, we sample

---

[11]There is one Spanish and German native speaker among the authors.

| Model | Acc | $\Delta_G$ | $\Delta_S$ |
|---|---|---|---|
| Flan-T5 | 65.1 | 7.2 | 35.1 |
| Flan-T5$_{\text{Few-Shot,Random}}$ | 68.9 | 4.1 | 22.2 |
| Flan-T5$_{\text{Few-Shot}}$ | **72.2**[*] | **2.1** | **19.6** |
| Flan-T5 | 67.5 | **2.3** | **-1.2** |
| Flan-T5$_{\text{Few-Shot,Random}}$ | 67.4 | 5.5 | 24.4 |
| Flan-T5$_{\text{Few-Shot}}$ | **69.8**[•] | **2.3** | -10.4 |

Table 5: Comparison of zero-shot and interpretability-guided few-shot debiasing with Flan-T5 in En-Es (top) and En-De (bottom). Best results per language in bold. [*] $: p \leq .01,$ [•]$p :\leq .05$

and translate N random examples from WinoMT (Random). For a fair comparison with the interpretability-guided sampling, we stratify random sampling too and build the prompts likewise.[12]

## 4.1 Results

Table 5 reports the comparison between Flan-T5 in zero-shot and using our debiasing approach. We report the best NT variants, which is NT-Female for Spanish and NT-Male for German. See Appendix D for full results.

In En-Es, choosing in-context examples guided by interpretability leads to a strong improvement in accuracy (+7.1), $\Delta_G$ (-5.1), and $\Delta_S$ (-15.5). Improvements in En-De are less marked, and $\Delta_S$ gets worse (+9.2 in absolute value). However, lower $\Delta_S$ might be artificially given by lower accuracy (Saunders and Byrne, 2020), as it happens for Flan-T5 zero-shot compared to Flan-T5$_{\text{Few-Shot}}$. Moreover, our approach leads to significant improvement over random sampling (see Appendix D.2 for details on significance).

Overall, these findings prove that interpretability scores, here $a_{pron,prof}$, can serve as a reliable signal to make fairer translations. We highlight how such improvements are enabled by a simple solution that requires no fine-tuning and only four human-written examples.

## 4.2 Qualitative Analysis

This section provides a qualitative analysis of the results obtained by Flan-T5$_{\text{Few-Shot}}$ in the En-Es setup, compared with the zero-shot Flan-T5.

Table 6 illustrates instances of wrong gender inflection in zero-shot translation (Flan-T5), contrasting them with the accurate inflection achieved

by Flan-T5$_{\text{Few-Shot}}$. In both stereotypical and non-stereotypical examples, we observe a correct shift in articles ("El" for male, "La" for female) and gender inflection corresponding to the profession (e.g., "the librarian" - "el bibliotecari**o**" (male), "la bibliotecari**a**" (female)). Interestingly, while Flan-T5 translates poorly the profession "clerk" with "el secretario" (second row), Flan-T5$_{\text{Few-Shot}}$ chooses the right word and gender inflection ("la empleada"). We attribute this improvement in translation to the presence of the profession "clerk" in the few-shot examples, which likely allows the model to learn the correct profession translation.

We also observe the behavior of the Flan-T5 and Flan-T5$_{\text{Few-Shot}}$ models across stereotypical and anti-stereotypical examples. Using the few-shot debiasing in Spanish, the model demonstrates a higher success rate in correcting the profession translations associated with anti-stereotypical examples (235) compared to stereotypical examples (80) out of a total of 348 identified examples.

Finally, we report results aggregated by profession in Table 7 for En-Es. By looking at the professions whose $\Delta_G$ changed the most, we see that `appraiser`, `therapist`, `sheriff`, `designer`, and `editor` achieved the most significant improvement in correct gender inflection with Flan-T5$_{\text{Few-Shot}}$. On the other hand, `veterinarian`, `teenager`, `resident`, `farmer`, and `advisor` worsened. I.e., despite showing overall fairer translations, improvements with Flan-T5$_{\text{Few-Shot}}$ are uneven across professions. This finding calls for more fine-grained inspections and overall dataset-level assessments.

## 5 Gender-Neutral Translation

To broaden our study and provide groundwork on interpretability for gender-neutral MT, we conducted a preliminary analysis of the 240 WinoMT samples requiring gender-neutral translation. These instances result from compiling templates with a gender-neutral pronoun (e.g., "The technician told the customer that *they* could pay with cash."). Table 10 provides a detailed overview of the results for Flan-T5 and mT0.

Considering Flan-T5 and En-Es, we did not find either inflected form in 67 (28%) cases (i.e., they did not match with any entry in our dictionary). From manual inspection, we observed that most cases contained either (1) an alternative valid form or synonym (e.g., "profesora" for "teacher" instead of "maestra") or (2) a semantically related but not

---
[12]Among the three possible configurations, i.e., Random-NT-Male, Random-NT-Female, and Random-NT-Random, we report here the latter.

| Text | Profession | Gender | Flan-T5 | Flan-T5_Few-Shot |
|---|---|---|---|---|
| **The librarian** helped the CEO to find the book because it is her job. | librarian | F | **El bibliotecario** ayudó al CEO a encontrar el libro porque es su trabajo. | **La bibliotecaria** ayudó al jefe a encontrar el libro porque es su trabajo. |
| The carpenter made a desk for **the clerk** because she needed a desk. | clerk | F | El carpintero hizo una mesa para **el secretario** porque necesitaba una mesa. | El carpentero hizo una mesa para **la empleada** porque necesitaba una mesa. |
| **The nurse** examined the farmer for injuries because he was caring. | nurse | M | **La enfermera** examinó al campesino para heridas porque era cuidadoso. | **El enfermero** examinó al campesino para heridas porque era cuidadoso. |
| **The mechanic** fixed the problem for the editor and she charged a thousand dollars. | mechanic | M | **El mecánico** resolvió el problema para el editor y le cobraron mil dólares. | **La mecánica** resolvió el problema para el editor y le cobrara mil dólares. |

Table 6: Gender misalignment examples in Flan-T5 En-Es translation vs. correct alignment in Flan-T5_Few-Shot. Two stereotypical (top) and two anti-stereotypical (bottom) examples are shown. Target profession in bold.

| Profession | Flan-T5 | Flan-T5_Few-Shot | $\Delta$ ($\downarrow$) |
|---|---|---|---|
| appraiser | 0.41 | 0.17 | -0.25 |
| therapist | 0.56 | 0.41 | -0.14 |
| sheriff | 0.52 | 0.45 | -0.06 |
| designer | 0.64 | 0.59 | -0.05 |
| editor | 0.79 | 0.77 | -0.03 |
| veterinarian | 0.17 | 0.56 | 0.39 |
| teenager | 0.17 | 0.56 | 0.39 |
| resident | 0.17 | 0.56 | 0.39 |
| advisor | 0.17 | 0.56 | 0.39 |
| farmer | 0.38 | 0.70 | 0.31 |

Table 7: Flan-T5 and Flan-T5_Few-Shot $\Delta_G$ performance across professions in En-Es and their difference ($\Delta$).

strictly correct translation – e.g., "pasajero" (eng: passenger) for "pedestrian" instead of "peatón". Interestingly, in some rarer cases, the model replaces the target profession with an indirect object pronoun (e.g., "The supervisor gave [the employee] feedback on their stellar performance." -> "El supervisión [le] dio información sobre su trabajo extraordinario."[13]). While these cases are a form of GNT, they come at the expense of information loss.

In the remaining matching 173 (72%) examples, the model used a male inflection 54% of the times but only 9% a feminine form, with the remaining cases not identified.[14] Some professions are exclusively inflected with feminine (e.g., nurse, receptionist) and others masculine (e.g., buyer, chef, chemist) forms. Moreover, we found that the importance given to the pronoun (they/them/their) when choosing a masculine form for the profession (i.e.,

$a_{pron,prof}$) is lower (median: 0.09) than in feminine cases (median: 0.16). Overall, these findings corroborate those on binary-gender instances because (1) Flan-T5 mostly defaults to masculine, overlooking the clue given by the pronoun, and (2) interpretability scores can be a valuable tool to detect such an unwanted behavior.

Whether we match it or not, the model inflects the pronoun's referent profession in most cases. Poor GNT capabilities of instruction-tuned models echo those recently found in commercial systems (Piergentili et al., 2023b). Similar results hold for mT0 (En-Es). In contrast, results on En-De show a sensibly lower number of matching instances (39% for Flan-T5, 35% for mT0). Through manual inspection, we attribute it mainly to using synonyms and wrong translations for Flan-T5 and failed translations for mT0. We report full results in Appendix C.3.

## 6 Related Work

**Gender Bias in MT.** As for other NLP tasks and models (e.g., Bolukbasi et al., 2016; Zhao et al., 2017; Rudinger et al., 2018, *inter alia*), the study of gender bias (Sun et al., 2019) has received much attention in MT. For a thorough review on the topic, we refer to Savoldi et al. (2021).

Most prominently, Stanovsky et al. (2019) presented the WinoMT data set for measuring occupational stereotypical gender bias in translations. Later, the data set was extended by Troles and Schmid (2021) for covering gender stereotypical adjectives and verbs. Prates et al. (2020) analyzed gender bias in Google Translate. Levy et al. (2021)

---

[13]Spanish "le" can translate to "to him/her."
[14]As per the WinoMT's official heuristic and morphological analysis tool.

focused on collecting natural data, while Gonen and Webster (2020) assessed gender issues in real-world input. As gender bias in MT is highly related to the typological features of the target languages, several recent studies focused on specific language pairs, e.g., English/Korean (Cho et al., 2019), English/Hindi (Ramesh et al., 2021), English/Turkish (Ciora et al., 2021), and English/Italian (Vanmassenhove and Monti, 2021). Daems and Hackenbuchner (2022) introduced a living community-driven collection of biased MT instances across many language pairs. Apart from data sets and methods for assessing gender bias in MT, researchers also proposed methods for bias mitigation. For instance, Escudé Font and Costa-jussà (2019) focused on embedding-based techniques (e.g., HardDebiasing), while Saunders et al. (2020) relied on gender inflection tags. Stafanovičs et al. (2020) analyze the effect word-level annotations containing information about subject's gender. Saunders and Byrne (2020) investigated the use of domain adaptation methods for bias mitigation, and Escolano et al. (2021) proposed to jointly learn the translation, the part-of-speech, and the gender of the target languages. Most recently, researchers have focused more on the role of gender-neutrality. As such, Lauscher et al. (2023) present a study on (neo)pronouns in commercial MT systems, and Piergentili et al. (2023a) propose more inclusive MT through generating gender-neutral translations.

**Interpretability for MT.** Previous work in context-aware MT (Voita et al., 2018; Kim et al., 2019; Yin et al., 2021; Sarti et al., 2023a) studied how models use (or do not use) context tokens by looking at internal components (e.g., attention heads, layer activations). More recent solutions enable the study of the impact of source *and* target tokens (Ferrando et al., 2022) or discover the causes of hallucinations (Dale et al., 2023). Concurrent work by Sarti et al. (2023b) uses post-hoc XAI methods to uncover gender bias in Turkish-English neural MT models. We expand their setup to more complex sentences and the notional-to-grammatical gender MT in two more languages.

**Mitigating Bias in Prompt-based Models.** Given that sufficiently large LMs exhibit increasingly good few-shot and zero-shot abilities (Brown et al., 2020a), researchers investigated variants of *prompting* as a particular promising technique. Relevant to us, several works (Sanh et al., 2022) pro-

posed to tune models for following instructions in multi-task learning regimes achieving surprisingly good task generalizability (e.g., Sanh et al., 2022; Chung et al., 2022, *inter alia*). However, also prompt-based models are prone to encode and amplify social biases. In this context, Lucy and Bamman (2021) showed that GPT-3 exhibits stereotypical gender bias in story generation. Schick et al. (2021) proposed self-diagnosis and self-debiasing for language models which they test on T5 and GPT-2. (Akyürek et al., 2022) investigated whether the form of a prompt, independent of the content, influences the measurable bias. In contrast, Prabhumoye et al. (2021) use instruction-tuned models to detect social biases in given texts.

In this work, we are the first to explore the use of interpretability scores for informing bias mitigation in instruction-tuned models, bridging the gap between fairness and transparency in MT.

## 7 Conclusion

This paper introduced the first extensive study on the evaluation and mitigation of gender bias in machine translation with instruction-tuned language models. Prominently, we studied the phenomenon through the lenses of interpretability and found that models systematically overlook lexical clues to inflect gender-marked words. Building on this finding, we proposed a simple and effective debiasing solution based on few-shot learning, where interpretability guides the selection of relevant exemplars.

## Acknowledgments

This project has in part received funding from Fondazione Cariplo (grant No. 2020-4288, MONICA) and the European Research Council (ERC) under the European Union's Horizon 2020 research and innovation program (No. 949944, INTEGRATOR). GA, FP, and DN are member of the MilaNLP group and the Data and Marketing Insights Unit of the Bocconi Institute for Data Science and Analysis. AL's work is funded under the Excellence Strategy of the German Federal Government and the Länder.

## Limitations

Our work comes with a number of limitations.

We chose German and Spanish as the translation target languages for our analysis. Our choice was motivated by (1) the presence of grammatical

gender in those languages, (2) their typological diversity, and (3) our access to native speakers of those languages for double-checking the models' translations and translating our few-shot examples. We know that none of these languages is resource-scarce and that including more languages would strengthen our study. However, given the depth of our study and our qualitative analyses, we leave expanding our findings to more languages for future research.

This work focuses on the standard WinoMT benchmark. Yet, the dataset is constructed by slot-filling templates, simplifying the analysis on several aspects (e.g., there is one gender-marking pronoun, only two possible referents, and simple sentence structure). However, we argue that our methodology will serve as essential groundwork for extensions on natural MT setups (e.g., Bentivogli et al., 2020; Currey et al., 2022).

Finally, we conduct the few-shot experiments only with the largest model variants. Including more model sizes here could lead to deeper insights. We chose to do so due to (1) the lousy translation quality we observed for smaller model sizes and (2) the computational effort associated with an environmental impact. We believe that our findings will generalize to other model sizes, providing a decent overall translation quality.

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

## A  A Note on Model Nomenclature

The terms *multitask prompted tuning* and *instruction finetuning*–or simply *instruction tuning*–refer to the same strategy of recasting existing NLP datasets using natural language and particular prompt-answer templates. Raffel et al. (2020) originally introduced instruction tuning as a complement of pre-training of text-to-text models on a variety of NLP tasks. More recently, instruction tuning has also been used to indicate the training process on human instructions, that lead to the rise of modern assistant LMs (e.g., Ouyang et al., 2022; Chiang et al., 2023; Dettmers et al., 2023, *inter alia*).

Although the two procedures have conflated, we can reasonably see recasted dataset (e.g., FLAN (Chung et al., 2022) or xP3) as instruction following data. Therefore, in this paper, we adopt the *instruction finetuning* nomenclature as a reasonble umbrella definition.

## B  Details on the Experimental Setup

### B.1  Interpretability

We use Integrated Gradient (IG) token-level attribution scores (Sundararajan et al., 2017). Following recent work on IG for NLP, we multiply raw scores

by input embeddings (Han et al., 2020) and integrate over 16 steps from the baseline. We run model inference pass using 8-bit quantization for efficiency (Dettmers et al., 2022). We set aggregation functions $f$ to take the highest absolute value score across the aggregated span, preserving the sign (e.g., $f([0.01, -0.3, 0.1]) = -0.3$), and $g$ to the Euclidean norm. We aggregate first over $f$ and then $g$ because we expect token-level per-unit scores to represent token attribution more expressively, and we do not want to lose such information with an initial pooling along the hidden size. We use Inseq (Sarti et al., 2023b) to compute and aggregate the scores.

### B.2  The WinoMT Corpus

WinoMT (Stanovsky et al., 2019) was created by combining the Winogender (Rudinger et al., 2018) and WinoBias (Zhao et al., 2018) coreference test sets. WinoMT consists of 3,888 instances, equally balanced between male and female genders and between stereotypical and non-stereotypical gender-occupation assignments.

Instances in WinoMT are constructed from templates where two professions interact in an arbitrary activity (e.g., "The `developer` argued with the `designer`"), and a pronoun, either personal or possessive, coreferent of one of the two professions ("because `she` did not like the `design`"). Two hundred forty corpus instances result from templates compiled using the pronouns they/their/them. They are intended to test gender-neutral translation capabilities.

### B.3  Translation with Baselines

We compare IFT models to two competitive baseline translation systems. We test Marian NMT models fine-tuned on the OPUS corpus,[15] matching the decoding strategy used for instruction models and no prompts. Moreover, we test GPT-3.5 using top p sampling (p=0.9, temperature=0.2, max_tokens=256) and prompt "Translate the following sentence into {tgt_lang}: {src_text}".

## C  Additional Results

### C.1  Gender Bias Evaluation

Table 8 reports full results on the gender bias evaluation of IFT models and several baselines. For Flan-T5, generally accuracy and fairness improve

---

[15] https://opus.nlpl.eu/

```
Q: Translate {src_text_few_shot} to {
    tgt_lang}?\n\n
A: {tgt_text_few_shot}\n\n\n
```

Figure 2: Few-shot debiasing prompt, individual exemplar template. `src_text_few_shot` and `tgt_text_few_shot` are human-translated parallel WinoMT instances.

with size. Flan-T5-Small is an exception, being the best model for $\Delta_S$ in En-Es. However, low accuracy might induce a very low $\Delta_S$ (Saunders and Byrne, 2020). Indeed, the same model is not able to produce meaningful results in En-De. Smaller mT0 models share this trait of low accuracy and unstable translations in $\Delta_S$.

## C.2 Impact of Decoding and Prompt Template

We conducted an extensive hyperparameter search on five decoding strategies and two prompt templates. We conducted all tests on the Europarl WMT' 06 test sets. We experimented with greedy search, beam search (n=4, no sampling), top k sampling (k = [5, 10, 20, 50, 80]), top p sampling (p = [0.4, 0.6, 0.8, 0.9, 0.95, 1.0] and temperature = [0.4, 0.7, 1]), and contrastive decoding (k = [2, 5, 10], and penalty alpha = 0.6). As for prompt templates, we used two from the FLAN collection, i.e., "{src_text} Translate this to {tgt_lang}?" and "Translate from {src_lang} to {tgt_lang}:\n\n{src_text}\n\n{tgt_lang}:".[16] We ran an exhaustive grid search on En-Es, identify the best setup, and translate En-De with the best setup found.

In line with prior literature, we found that beam search (n=4, no sampling) is the best decoding strategy across all languages, models, and model sizes. Moreover, Figure 3 and 4 show a clear increasing trend in performance as models get bigger, with mT0 being slightly better than Flan-T5 on average. Notably, only the largest models achieve positive quality estimation scores (COMET20), even though non comparable to those of supervised models. Table 9 reports all results on both En-Es and En-De for all tested models.

## C.3 Gender-Neutral Translation

Table 10 reports a complete overview of the results on the 240 gender-neutral instances for Flan-T5-

---

[16]https://github.com/google-research/FLAN/blob/main/flan/v2/templates.py

XXL and mT0-XXL. We observe several interesting findings. (1) Both models tend to inflect into masculine or feminine forms more frequently when translating into Spanish (avg: 75%) than German (avg: 37%). (2) Frequent gender inflections – despite neutral pronoun constructions – underscore models' poor capabilities in this task, as they rarely resort to gender-neutral translations. (3) When inflecting the gender, both models use masculine substantially more often than feminine, showing a persistent gender bias even in GNT. (4) $a_{pron,prof}$ for masculine translations is lower than that for feminine ones across both languages and models. This finding supports the intuition that models "overlook" overt contextual clues to inflect gender and default to male forms.

Since En-De has a sensibly lower number of matching instance, we manually inspected them looking for the causes. Table 11 reports full statistics. Flan-T5 omitted the target phrase once; mT0 did the same twice and used a gender neutral plural once. Notably, while Flan-T5 mainly uses synonyms or translates profession words wrongly, mT0 fails many translations by verbatim copying the input sentence to the output.

## D   Details on the Debiasing

Table 12 reports the full list of results with different strategies for few-shot prompting. In three cases out of four (En-Es Flan, En-De Flan, and En-Es mT0), the proposed solution for few-shot examples selection leads to better accuracy with at least one variant of the non-target profession. In En-Es, for both Flan-T5 and mT0, the best version is `Few-Shot NT-Female` (i.e., providing examples where the non-target profession is feminine-inflected).

We do not report the results for mT0 En-De as the model is not able to handle any few-shot template in this setup. The resulting generation are almost always in a language different than the target one or they are empty.

## D.1   Few-Shot Prompt

We reuse a FLAN few-shot template for our few-shot debiasing approach. Specifically, we concatenate N=4 times the exemplar template reported in Figure 2.

| Model | Spanish | | | German | | |
|---|---|---|---|---|---|---|
| | Acc | $\Delta_G$ | $\Delta_S$ | Acc | $\Delta_G$ | $\Delta_S$ |
| Google Translate* | 53.1 | 23.4 | 21.3 | 59.4 | 12.5 | 12.5 |
| Microsoft Translator* | 47.3 | 36.8 | 23.2 | **74.1** | **0.0** | 30.2 |
| Amazon Translate* | 59.4 | 15.4 | 22.3 | 62.4 | 12.0 | 16.7 |
| Marian NMT | 56.8 | 16.9 | 19.7 | 62.0 | 9.9 | 15.2 |
| GPT-3.5 | 55.2 | 23.1 | 48.5 | 48.3 | 25.2 | 24.6 |
| Flan-T5-Small | 44.4 | 38.5 | **8.7** | ** | ** | ** |
| Flan-T5-Base | 47.7 | 31.5 | 22.8 | 39.5 | -17.8 | -18.7 |
| Flan-T5-Large | 51.7 | 17.1 | 44.7 | 52.3 | -4.9 | -5.9 |
| Flan-T5-XL | 58.5 | 16.0 | 40.3 | 68.0 | 2.39 | 14.3 |
| Flan-T5-XXL | **65.6** | **6.7** | 34.3 | 67.5 | 2.3 | **-1.2** |
| mT0-Small | 40.6 | 17.9 | 13.4 | 46.9 | 59.0 | ** |
| mT0-Base | 54.2 | 12.2 | 27.4 | 47.0 | 59.0 | ** |
| mT0-Large | 61.5 | 5.8 | 26.7 | 46.9 | 59.0 | ** |
| mT0-XL | 51.0 | 29.6 | 51.6 | 47.6 | 54.3 | 8.2 |
| mT0-XXL | 51.9 | 29.7 | 33.8 | 56.0 | 24.2 | 21.1 |

Table 8: Gender bias evaluation on WinoMT on all tested models. Instruction-tuned models with beam search (n=4, no sampling) decoding. Best models in bold. *Results reported from Stanovsky et al. (2019). **Results not reported due to unstable translations (i.e., mostly empty or not in the target language).

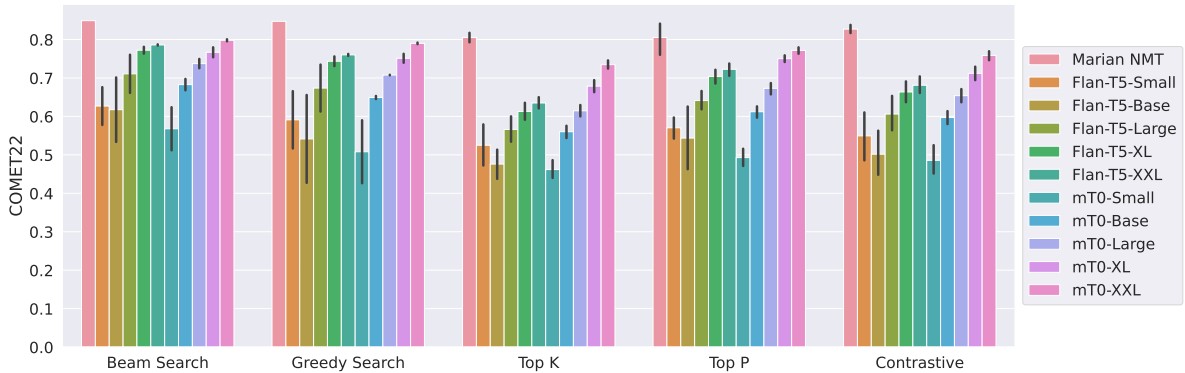

Figure 3: Flan-T5 and mT0 zero-shot Translation performance (reference-based COMET-22) for different decoding and model size on the Europarl WMT' 06 En-Es test set. Marian NMT supervised model for comparison.

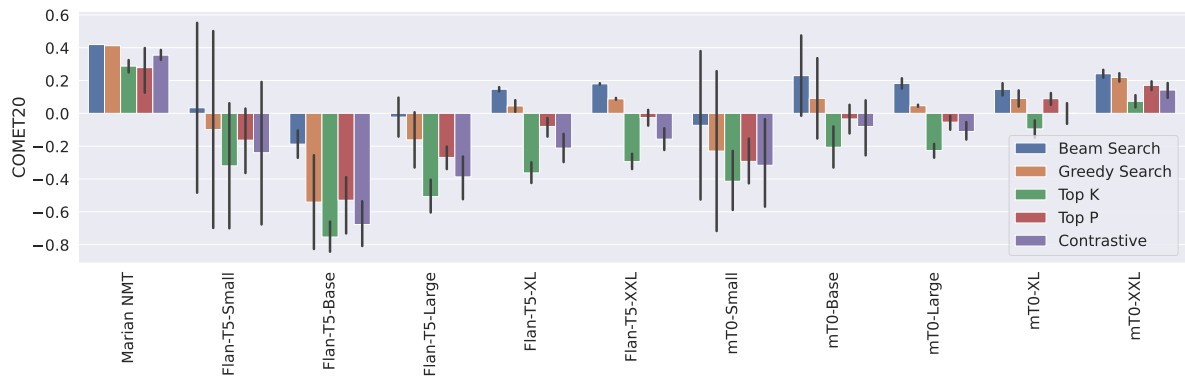

Figure 4: Flan-T5 and mT0 zero-shot Translation performance (reference-free COMET-20) for different decoding and model sizes on the Europarl WMT' 06 En-Es test set. Marian NMT supervised model for comparison.

| Model | COMET-22 | COMET-20 | BERTScore | BLEU-2 | BLEU-4 |
|---|---|---|---|---|---|
| Marian NMT | *0.85* | *0.42* | *0.89* | *0.55* | *0.39* |
| Flan-T5-Small | 0.58 | -0.48 | 0.81 | 0.31 | 0.16 |
| Flan-T5-Base | 0.70 | -0.10 | 0.84 | 0.39 | 0.23 |
| Flan-T5-Large | 0.76 | 0.10 | 0.85 | 0.44 | 0.27 |
| Flan-T5-XL | 0.78 | 0.16 | **0.86** | **0.45** | **0.29** |
| Flan-T5-XXL | 0.79 | 0.18 | **0.86** | **0.45** | **0.29** |
| mT0-Small | 0.51 | -0.53 | 0.73 | 0.14 | 0.05 |
| mT0-Base | 0.70 | -0.02 | 0.81 | 0.29 | 0.15 |
| mT0-Large | 0.75 | 0.15 | 0.82 | 0.32 | 0.18 |
| mT0-XL | 0.78 | 0.18 | 0.81 | 0.34 | 0.21 |
| mT0-XXL | **0.80** | **0.27** | 0.82 | 0.42 | 0.27 |
| Marian NMT | *0.80* | *0.43* | *0.85* | 0.42 | *0.26* |
| Flan-T5-Small | 0.52 | -0.51 | 0.76 | 0.18 | 0.07 |
| Flan-T5-Base | 0.63 | -0.12 | 0.80 | 0.26 | 0.12 |
| Flan-T5-Large | 0.70 | 0.15 | 0.82 | 0.32 | 0.17 |
| Flan-T5-XL | 0.73 | 0.26 | 0.82 | 0.32 | 0.17 |
| Flan-T5-XXL | 0.75 | 0.31 | 0.82 | *0.82* | 0.18 |
| mT0-Small | 0.44 | 0.02 | 0.70 | 0.01 | 0.00 |
| mT0-Base | 0.54 | 0.39 | 0.75 | 0.04 | 0.01 |
| mT0-Large | 0.53 | 0.33 | 0.74 | 0.03 | 0.01 |
| mT0-XL | 0.60 | 0.32 | 0.77 | 0.10 | 0.04 |
| mT0-XXL | 0.75 | 0.42 | 0.81 | 0.28 | 0.15 |

Table 9: Zero-shot performance for the best configuration of decoding and prompt template on Europarl WMT' 06 En-Es (top) and En-De (bottom) test sets. Bold and italic indicate best instruction-tuned and overall models per target language, respectively. Supervised Marian NMT baselines (top rows) for comparison.

| | Spanish | | German | |
|---|---|---|---|---|
| **Translated Gender** | **Matching (%)** | $a_{pron,prof}$ | **Matching (%)** | $a_{pron,prof}$ |
| Female | 7 | 0.1588 | 11 | 0.1765 |
| Male | 39 | 0.0892 | 27 | 0.1362 |
| Neutral/Unknown | 26 | 0.1510 | 1 | 0.2134 |
| Non-Matching | 28 | - | 61 | - |
| Female | 5 | 0.3565 | 4 | 0.2996 |
| Male | 50 | 0.2008 | 30 | 0.2021 |
| Neutral/Unknown | 23 | 0.1863 | 1 | 0.3722 |
| Non-Matching | 22 | - | 65 | - |

Table 10: Statistics on gender-neutral cases in WinoMT for Flan-T5-XXL (top) and mT0-XXL (bottom). Number of matching occurrences of female- or male- inflected pronoun referents, and median $a_{pron,prof}$.

## D.2 Statistical Significance

We compute statistical significance of the difference in performance between few-shot with random sampling of examples and choosing using $a_{prof,prof}$. We use bootstrap sampling (Søgaard et al., 2014, n=1000, sample size-30%). Interpretability informed sampling is better than random sampling in terms of macro F1 score ($p \leq .01$, in En-Es and En-De), and accuracy ($p \leq .01$ in En-Es, $p \leq .05$ in En-De).

| Reason | Flan-T5 | mT0 |
|---|---|---|
| Synonym (male) | 34.3 | 23.4 |
| Wrong Translation | 21 | 0 |
| Grammatical inflection (male) | 18.2 | 16.2 |
| Synonym (female) | 10.5 | 2.6 |
| Grammatical inflection (neutral) | 8.4 | 16.2 |
| Synonym (neutral) | 4.9 | 3.2 |
| English Translation | 2.1 | 36.4 |
| Phrase omitted | 0.7 | 1.3 |
| Gender Neutral Plural | 0 | 0.6 |

Table 11: Frequency (%) of each cause in non-matching instances in gender-neutral translation for En-De.

## E  Carbon Footprint

Experiments were conducted using a private infrastructure, which has a carbon efficiency of 0.29 $kgCO_2eq$/kWh. A cumulative of 184 hours (53 for translation, 14 for evaluation, 117 for generating feature attribution scores) of computation was performed on hardware of type A100 PCIe 80GB (TDP of 250W). Total emissions are estimated to be 13.34 $kgCO_2eq$, none of which were directly offset. Estimations were conducted using the Machine-Learning Impact calculator presented in (Lacoste et al., 2019).

## F  Release of Data Artifacts

We release code to reproduce our experiments at https://github.com/MilaNLProc/interpretability-mt-gender-bias. Moreover, we plan to release all data artifacts produced in our study hoping to foster future research in the field, including (1) integrated gradient scores, (2) human-refined GPT-3.5 translation of WinoMT professions, and (3) human-translated seed demonstrations for few-shot learning. We will release any additional content not listed here in the paper repository.

| Model | Demonstration Sampling | Spanish | | | German | | |
|---|---|---|---|---|---|---|---|
| | | Acc | $\Delta_G$ | $\Delta_S$ | Acc | $\Delta_G$ | $\Delta_S$ |
| Flan-T5-XXL | - | 65.6 | 6.7 | 34.3 | 67.5 | **2.3** | -1.2 |
| Flan-T5-XXL$_{\text{Few-Shot}}$ | Random | 68.9 | 4.10 | 22.2 | 67.4 | 5.5 | 24.4 |
| Flan-T5-XXL$_{\text{Few-Shot}}$ | NT-Female | **72.2** | **2.1** | 19.6 | 65.1 | -6.1 | -8.2 |
| Flan-T5-XXL$_{\text{Few-Shot}}$ | NT-Male | 70.3 | 3.4 | 24.1 | **69.8** | **2.3** | -10.4 |
| Flan-T5-XXL$_{\text{Few-Shot}}$ | NT-Random | 68.7 | 4.4 | 19.9 | 64.5 | -6.8 | -7.4 |
| mT0-XXL | - | 51.9 | 29.7 | 33.8 | 56.0 | 24.2 | 21.1 |
| mT0-XXL$_{\text{Few-Shot}}$ | Random | 46.6 | 25.1 | **11.7** | ** | ** | ** |
| mT0-XXL$_{\text{Few-Shot}}$ | NT-Female | 60.8 | 9.9 | 21.1 | ** | ** | ** |
| mT0-XXL$_{\text{Few-Shot}}$ | NT-Male | 58.2 | 16.0 | 31.0 | ** | ** | ** |
| mT0-XXL$_{\text{Few-Shot}}$ | NT-Random | 60.6 | 11.9 | 26.5 | ** | ** | ** |

Table 12: Gender bias evaluation on WinoMT for different strategies of human-written demonstration sampling. Zero-shot variants in top rows. Best results in bold. **Results not reported due to unstable translations (i.e., mostly empty or not in the target language).