# OpenReview forum: "A Tale of Pronouns: Interpretability Informs Gender Bias Mitigation for Fairer Instruction-Tuned Machine Translation"
_EMNLP/2023/Conference — EMNLP 2023 Main_

### Official Review · Reviewer_HKEt · 2023-08-01

**Typos Grammar Style And Presentation Improvements:** 35-36
**Soundness:** 4

**Excitement:**

4: Strong: This paper deepens the understanding of some phenomenon or lowers the barriers to an existing research direction.

**Missing References:**

- Gender Bias in MT + Interpretability: The demo paper associated to the Inseq library (https://aclanthology.org/2023.acl-demo.40/) you employed contains a case study on gender bias in machine translation that leverages the same techniques you used, and which should be cited as previous/concurrent work showing how feature attribution methods can be used to identify gender bias in machine translation settings.
- Gender Bias in MT + Interpretability: Previous work in context-aware MT (https://aclanthology.org/2021.acl-long.65.pdf#cite.voita-etal-2018-context, https://aclanthology.org/2021.acl-long.65/, https://aclanthology.org/D19-6503/) focused on the translation of texts containing inter-sentential gendered pronoun coreference, finding that the cue gendered word in the context is not attended adequately during translation. This finding seems very related to the overlooking of gendered pronouns shown in this work.
- Other MT Bias + Interpretability: Recent work use similar feature attribution techniques on MT models to identify toxic generations (https://arxiv.org/abs/2210.03070) and hallucinations (https://aclanthology.org/2023.acl-long.3/). While not strictly related to gender bias, these seem relevant to mention given the match in area of application and methods used.
- Mitigating Bias in Prompt-based Models: PALMS (https://proceedings.neurips.cc/paper/2021/hash/2e855f9489df0712b4bd8ea9e2848c5a-Abstract.html) and more recently LIMA (https://arxiv.org/abs/2305.11206) are two seminal works addressing the debiasing/alignment of prompted language models thanks to a small representative set of value-aligned examples.
- Mitigating Bias in Prompt-based Models: Related to your Interpretability-guided Debiasing for prompted LMs, RAMP (https://aclanthology.org/2023.acl-short.126/) is a sampling procedure for in-context examples selection to improve the style accuracy of LMs prompted for gender-controlled MT.

**Paper Topic And Main Contributions:**

The paper aims to leverage established post-hoc interpretability techniques to study and debias gender bias in instruction-tuned models for the machine translation task. Through several experiments on WinoMT En -> De and En -> Es, authors showcase the gender bias in translations produced by recent instruction-tuned multilingual models. By leveraging importance scores obtained during the generation of gendered occupation terms, authors highlight the propensity of these models to overlook the gendered pronouns in the source. Finally, a sampling procedure to select in-context examples from the importance given to pronouns is introduced, showing significant improvements in debiasing translations compared to baseline results.

**Questions For The Authors:**

A. The example of Figure 1 showcases an ambiguous coreference that humans can resolve employing semantic grounded information (i.e. it is usually the person that wins money that buys gifts for others, and not vice-versa). While in the example the feminine gender is missing regardless of the choice of coreferent, do you think that even a debiased MT model would be able to perform the coreference correctly, in the absence of further information that we usually may give for granted? This could be potentially related to the large accuracy gap between NT-Female and NT-Random results in Table 10.

B. Why is aggregation of raw attribution scores performed as $A = g(f(A_r))$ instead of $A = f(g(A_r))$ i.e. why is subword aggregation performed before aggregating per-unit scores at a token level? I suspect the reason to be the expressivity of the resulting vectors, but if this is the case it would need to be clearly mentioned at least in a footnote/appendix.

C. If the Euclidean norm is used to obtain per-token attribution scores, these are strictly positive. In light of this, are scores normalized across input words, such that $\sum_{s \in S} a_s = 1,  \forall a_s \in A_s = \mathbb{R}^S$? If this is the case, $a$ values from Table 3 can be interpreted as the relative % of attention to $ctrl, prof$ and $pron$ respectively when generating the target occupation term, which would be preferable to simplify understanding of the results.

D. Were more modern alignment approaches, such as AWESOME (https://github.com/neulab/awesome-align), tested alongside FastAlign? The improved quality of those might have saved quite some effort with lexical translation and matching.

E. You mention that the instruction-tuned models were provided with MT tuning examples in xP3 and FLAN. It is not clear to me whether the same template adopted by these datasets was used, given that it would be the most natural choice to ensure in-domain prompts.If this is not the case, could you justify your choice of using another MT prompt, possibly harming model performance?

**Reasons To Accept:**

The paper is well-written and showcases an interesting and timely application of post-hoc interpretability methods to support the analysis and reduction of gender bias in established generative language models. The authors provide an extensive empirical evaluation of several large-scale language models and generation settings while also analyzing more granular failure cases of these models from a qualitative standpoint. To my knowledge, this paper is one of the first to show a direct application of interpretability techniques that could mitigate representational biases in deployed MT systems. Moreover, the proposed method to reduce gender bias is simple but effective and suggests the potential of leveraging such techniques for debiasing other natural language generation tasks.

**Reasons To Reject:**

While results are generally promising, some parts of the paper contain strong statements/conclusions but sometimes lack further clarifications or explicit supporting evidence. I refer to these more explicitly in the "Typos Grammar Style And Presentation Improvements" section. Importantly, these are limited to some sections and in my opinion do not compromise the overall validity and quality of the work.

More generally, I am not quite convinced by the authors' assumption that results of Section 4 "motivate their [instruction-tuned LMs] adoption in real use cases", given the best instruction-tuned model with 11B parameters, (requiring significant resources for inference) cannot perform on par with a small (60M?) Marian NMT model trained for the task. In scenarios where the usage of these models is constrained to translation, the choice of the latter would be preferable. In light of this, the benefits of less-performant multi-purpose systems like the ones employed in this study should be discussed more explicitly to convince the reader of their relevance.

**Reproducibility:**

5: Could easily reproduce the results.

**Reviewer Confidence:**

4: Quite sure. I tried to check the important points carefully. It's unlikely, though conceivable, that I missed something that should affect my ratings.

---

> ### Author Rebuttal · Authors · 2023-08-27
>
> Thank you for your detailed feedback and for recognizing our work's novelty. Let us address your comments and Questions.
>
> **Real-World Adoption of Instruction Fine-tuned LMs.** We agree that supervised (smaller) LMs still outperform larger Instruction Fine-tuned (IFT) LMs. However, studying biases and applications of the latter is highly timely for two main reasons. First, pressing evidence shows that IFT LMs are narrowing the gap against task-specific supervised models (e.g., Flan-T5, Chung et al., 2022; GPT3.5, and GPT4, Ouyang et al., 2022; PaLM, Chowdhery et al., 2022) in many NLP tasks, mainly due to larger scale or training sources (for a reference, see Table 9 and Figure 2, where translation quality improves as models get larger). Hence, we believe IFT models will match or surpass task-supervised MT LMs. Second, IFT models will become widespread (if not already) as they are more accessible with easy-to-use UI (e.g., ChatGPT).
> We will include and deepen this discussion in the main body upon acceptance.
>
> **(Q-A)** We agree with your point, which is of extreme interest. Indeed, the WinoMT suite exposes models to this ambiguity with most samples. Take, for instance, “The mover outran the hairdresser because she was in shape.” Here, common sense would help us relate the action (outrunning) to the explanation (being in good shape) to identify what is the antecedent of “she” (all of this even before considering any stereotypical association on roles). As the WinoMT examples do not let us disentangle coreference capabilities from stereotypical resolution, it is hard to tell what debiasing models look at (e.g., are NT-Female few shots helpful because the model has at least one feminine form in the sentence?). We believe this is an exciting research direction we will explore in future work.
>
> **(Q-B)** That is correct; we expect token-level per-unit scores to represent token attribution more expressively, and we do not want to lose that information with an initial hidden pooling. We will discuss this further in the paper.
>
> **(Q-C)** We do not normalize scores across input words. We thought about that and decided not to pursue this way because non-normalized scores do not depend on the input length and can be compared across samples.
>
> **(Q-D)** We did not test other alignment methods, but thank you for pointing this out. We will consider AWESOME for future work.
>
> **(Q-E)** We used unmodified templates from the FLAN collection. This detail is reported in Appendix D.1.
>
> **Missing References.** Thank you for pointing out additional related work. We will further discuss them in the main body upon acceptance.
>
> **Comments on Writing and Clarifications.** Thank you for the detailed feedback. We will clarify each point in the revised paper.
> - 35-36, 182-185, 196-197, Figure 1, Appendix A, Table 9: We agree with the suggested changes and will implement them.
> - Appendix B: We will discuss that given the chosen f and g, we have token attribution scores that are strictly positive.
> - 231: We manually checked for grammatical errors and fixed them if present. We will clarify it in the paper.
> - 288-289: Yes, they correspond to the XXL variants; we will specify it in Table 1.
> - 294-298: We will fix the typo and clarify that the COMET-20 automatic metric is a reference-free neural quality estimation model whose output reflects human-generated quality scores used for training.
> - 351-357: In 3 out of 4 cases (datasets+models), the most accurate class is associated with the highest a_prof,prof values. However, as you noticed, there is no clear correlation with all the accuracy and a_prof,prof values. We will re-write the sentence to clarify it better.
> - 396-403: The findings were indeed obtained by looking at the percentage difference and checked with a correlation test; we will provide values to support the sentence.

---

### Official Review · Reviewer_eP6E · 2023-08-04

**Soundness:** 3

**Excitement:**

4: Strong: This paper deepens the understanding of some phenomenon or lowers the barriers to an existing research direction.

**Paper Topic And Main Contributions:**

The paper addresses gender bias in pretrained language models from the standpoint of interpretability. It explores the source of gender-based mistakes, and uses the findings to motivate a prompt-based mitigation method.

**Questions For The Authors:**

The paper states early in section 1 that gender bias in pretrained LMs puts "their societal beneficial use at risk" - this is not directly related to the paper's approach, but it was an unusual phrase and I wondered what was meant by it?

**Reasons To Accept:**

- Taking an explainability approach towards the gender translation problem is novel and interesting, as well as having the potential to guide future work on the problem by making it clearer what the problem is.

- Section 6 describes an interesting experiment. It seems from the discussion that there was some overlap between the professions in the few-shot-example-list, and the professions to translate in WinoMT. If so, it would be interesting to know how much of an impact that overlap had, perhaps by choosing a non-profession-overlapping list for each input. But regardless the experiment is interesting to read.

- The paper is for the most part clearly written (5.2 a little hard to follow), and the results clearly laid out and understandable. Great to see both numerical and qualitative analysis also.

- There was plenty of detail given for reproducibility purposes, and especially good to see scripts attached to reproduce the work

**Reasons To Reject:**

- The paper claims that there are insufficient resources and data for inquiries on non-binary and gender-neutral MT. It makes use of WinoMT (2019) - an add-on to WinoMT, including an adaptation dataset, to extend to non-binary translation from English into Spanish and German was made available in 2020 (Saunders, Sallis & Byrne, which is already cited by this paper). The paper would be stronger if it made use of the resources, or offered an explanation for why the existing resources are unsuitable. An alternative would be to simply try translating such sentences with gender-neutral pronouns in the source, and qualitatively examine even a few of the results, similar to Table 6.

Edit to add: The authors have responded with some interesting preliminary notes on the neutral-pronoun sentences in WinoMT. Some analysis along these lines would be a great addition to the paper if accepted.

**Reproducibility:**

4: Could mostly reproduce the results, but there may be some variation because of sample variance or minor variations in their interpretation of the protocol or method.

**Reviewer Confidence:**

4: Quite sure. I tried to check the important points carefully. It's unlikely, though conceivable, that I missed something that should affect my ratings.

**Typos Grammar Style And Presentation Improvements:**

- The experiments in 5.2, while interesting, were a little difficult for me to follow. Perhaps it would be clearer if it were broken up a little, say into self-contained sections for the meaning and findings of each of a_prof,prof, a_pron,prof, etc.

---

> ### Author Rebuttal · Authors · 2023-08-27
>
> Thank you for your insightful comments and for recognizing the novelty of the approach and the potential insights it can provide. We appreciate you found the paper well-written, and we will further revise the writing in the final draft following your suggestions. Let us address the points you have raised as Weaknesses (W) and reply to your Questions (Q).
>
> **(W) Inquiries in Gender-Neutral Translations.** We believe observing the models' behavior in gender-neutral cases is indeed of extreme interest, and it would require a quite careful and extensive analysis from the interpretability angle (e.g., different forms of gender-neutral translations might exist when translating to a grammatical gender language, and each of them might require different observations and considerations). While this may require a specific research effort, we will provide in the paper some preliminary considerations based on the interpretability scores on the 240 gender-neutral examples in WinoMT.
>
> Considering Flan-T5-XXL and En-Es, we found the target profession in 173 (72%) examples. Among these, the model used a male inflection 54% of the time but only 9% a feminine form, with the remaining cases not identified. Some professions are exclusively inflected with feminine (nurse, receptionist, victim) and others masculine (buyer, chef, chemist). Moreover, we found that the importance given to the pronoun (they/them/their) when choosing a masculine form for the profession (i.e., a_{prof,pron}) is lower (median: 0.9) than in feminine cases (median: 0.16).
> These findings corroborate those in our paper because 1) Flan-T5 uses the masculine form by default, overlooking the clue given by the pronoun, and 2) interpretability can be a valuable tool to detect such an unwanted model behavior.
>
> **(W) Overlap between Professions in the Prompt and Examples.** Our proposed debiasing uses the same examples (i.e., in-context “shots”) for translating every sample. By doing so, the average overlap across the 3,888 WinoMT set is close to none (very few examples out of the total will contain one of the professions mentioned in the four examples we sample to build the debiasing prompt).
> Moreover, we thought about using different shots for every input; however, that would entail manually translating more examples (in the worst case, one-fourth of the dataset), whereas we show that we can reduce bias and improve performance with as little as translating 4.
>
> **(Q) Societal Beneficial Use at Risk.** Instruction Fine-tuned models are promising tools for many language tasks; hence, they can have a highly beneficial impact on society. However, once a model learns societal biases and unfair stereotypes, it risks perpetrating them at scale. Failing to evaluate and mitigate such biases puts their use at risk because it will be unsafe to deploy and let people use them.

---

### Official Review · Reviewer_6GEX · 2023-08-05

**Soundness:** 3

**Excitement:**

4: Strong: This paper deepens the understanding of some phenomenon or lowers the barriers to an existing research direction.

**Paper Topic And Main Contributions:**

This paper investigates the gender bias issue of instruction-finetuned models in machine translation task. They use a standard WinoMT benchmark to evaluate gender bias of two instruction-finetuned models, Flan-T5 and mT0, in zero/few-shot settings. They also provide further analysis and bias mitigation methods based on existing interpretability methods. The main contributions are as follows:
- while many work cover gender bias on supervised MT models, this work specifically focuses on instruction-tuned models and its susceptibility to biases.
- it uses integrated gradients (IG) interpretability-based method to analyze how model-internal effects lead to gender bias.
- it proposes a bias mitigation method based on few-shot prompting using the interpretability scores.

**Questions For The Authors:**

A. Results from Table 10 in Appendix indicate that the inflective form of NT (profession whose gender is unknown) affects performance considerably. While all the NT variants for interpretability-guided debiasing is reported and the best variant is included in the main paper, it is unclear whether all NT variants are considered for the baseline method of random sampling. Which version is used?

B. In L458, the baseline model having lower $\Delta_S$ of -1.2 compared to proposed method's -10.4 is explained that it might be artificially lower due to its lower accuracy. However, I'm wondering whether this is possible since  $\Delta_S$ represents the accuracy difference between stereotypical and anti-stereotypical samples, and while the accuracy difference between baseline and proposed is 69.8-67.5=2.3, $\Delta_S$ difference is -10.4-(-1.2)=-9.2, and thus much bigger.

**Reasons To Accept:**

Strengths:
- this paper investigates the well-defined MT gender bias problem for the instruction-tuned models and few-shot prompting, which is being rapidly adopted and used in recent NLP works. As the authors state, the gender bias problem of these few-shot setup is relatively not well known.
- Their proposed method of using integrated gradients based attribution scores to  further analyze gender bias problems provides interesting insights.

**Reasons To Reject:**

Weaknesses:
- The proposed bias mitigation method makes use of sampling from WinoMT dataset to use as few-shot examples, and then proceeds to use the same set for gender bias evaluation (Table 5). This raises a question of whether the proposed method is especially effective for only the WinoMT dataset, or is equally effective and generalizable to other sentences of different styles.
- Previous works on gender debiasing note a trade-off between translation quality and bias reduction [1]. While the gender debiasing effect of the proposed method has been measured, its effect on translation performance is unclear.

[1]  Saunders and Bryne, Reducing Gender Bias in Neural Machine Translation as a Domain Adaptation Problem, ACL 2020.

**Reproducibility:**

4: Could mostly reproduce the results, but there may be some variation because of sample variance or minor variations in their interpretation of the protocol or method.

**Reviewer Confidence:**

4: Quite sure. I tried to check the important points carefully. It's unlikely, though conceivable, that I missed something that should affect my ratings.

---

> ### Author Rebuttal · Authors · 2023-08-27
>
> Thank you for your helpful feedback. Let us address the points you have raised as Weaknesses (W) and reply to your Questions (Q).
>
> **(W) Generalization to Different Styles.** Our paper lays the foundation for linking interpretability and occupational gender bias on recent instruction fine-tuned models, proposing a mitigation strategy. We used WinoMT because it is an established benchmark for studying this issue, and it also provides several meaningful markers (e.g., the source pronoun and target profession) that helped us perform the analysis. Generalization to different domains and styles would require labeling these markers on new datasets or finding different ones, which is beyond the scope of this paper. We agree that exploring more complex setups, e.g., natural sentences like MT-GenEval (Currey et al., 2022), is an exciting research we will explore.
>
> **(W) Bias Reduction vs. Translation Performance.** Table 5 reports improved performance on correct gender inflection (Accuracy column), which we can loosely relate to an on-par or better overall translation quality after debiasing. We cannot measure translation quality automatically as WinoMT does not provide gold translations. Moreover, our solution uses in-context learning and cannot quickly adapt to out-of-domain data, as in Saunders and Bryne, 2020.
>
> **(Q-A)** For the random baseline, we randomly sample a female- or male-inflected Non-Target target occupation (i.e., equivalent to NT-Random in Table 10), keeping the stratification over target gender and stereotype type. Thank you for catching this missing detail; we will include it in the main body.
>
> **(Q-B)** As reported by Saunders and Byrne, 2020, Delta_S can be heavily skewed by low accuracy: imagine a system using the male form for each sample. Since WinoMT has the same number of stereotypical and anti-stereotypical instances, Delta_S will be 0 (best value) as accuracy over the two sets will be identical. We hence believe Flan-T5 is outscoring the debiased variant on Delta_S because of that.

---

### Meta-Review · Area_Chair_DrLs · 2023-09-19

**Recommendation:** 4

**Metareview:**

The paper "A Tale of Pronouns: Interpretability Informs Gender Bias Mitigation for Fairer Instruction-Tuned Machine Translation" presents work on examining how gender is translated from languages not having grammatical gender to languages having grammatical gender. To this end, the paper presents work on computing the gender bias in translated data. The paper also proposes a method for reducing the bias in translations.

The main criticism points mentioned by the reviewers refers to previous work, but also to the generalization of the results based on the used data set to general MT.
In support of the paper the reviewers mention the experimental setup, the presented details and the insights gained from the experiments.

---

### Decision · Program_Chairs · 2023-10-07

**Decision:**

Accept-Main

**Comment:**

The paper "A Tale of Pronouns: Interpretability Informs Gender Bias Mitigation for Fairer Instruction-Tuned Machine Translation" presents work on examining how gender is translated from languages not having grammatical gender to languages having grammatical gender. To this end, the paper presents work on computing the gender bias in translated data. The paper also proposes a method for reducing the bias in translations.

The main criticism points mentioned by the reviewers refers to previous work, but also to the generalization of the results based on the used data set to general MT.
In support of the paper the reviewers mention the experimental setup, the presented details and the insights gained from the experiments.